# Deciphering the Functions of Raphe–Hippocampal Serotonergic and Glutamatergic Circuits and Their Deficits in Alzheimer’s Disease

**DOI:** 10.3390/ijms26031234

**Published:** 2025-01-30

**Authors:** Wanting Yu, Ruonan Zhang, Aohan Zhang, Yufei Mei

**Affiliations:** Hubei Clinical Research Center for Alzheimer’s Disease, Brain Science and Advanced Technology Institute, School of Medicine, Wuhan University of Science and Technology, Wuhan 430065, China

**Keywords:** serotonin, hippocampus, glutamatergic, Alzheimer’s disease, cognition, raphe

## Abstract

Subcortical innervation of the hippocampus by the raphe nucleus is essential for emotional and cognitive control. The two major afferents from raphe to hippocampus originate from serotonergic and glutamatergic neurons, of which the serotonergic control of hippocampal inhibitory network, theta activity, and synaptic plasticity have been extensively explored in the growing body of literature, whereas those of glutamatergic circuits have received little attention. Notably, both serotonergic and glutamatergic circuits between raphe and hippocampus are disrupted in Alzheimer’s disease (AD), which may contribute to initiation and progression of behavioral and psychological symptoms of dementia. Thus, deciphering the mechanism underlying abnormal raphe–hippocampal circuits in AD is crucial to prevent dementia-associated emotional and cognitive symptoms. In this review, we summarize the anatomical, neurochemical, and electrophysiological diversity of raphe nuclei as well as the architecture of raphe–hippocampal circuitry. We then elucidate subcortical control of hippocampal activity by raphe nuclei and their role in regulation of emotion and cognition. Additionally, we present an overview of disrupted raphe–hippocampal circuits in AD pathogenesis and analyze the available therapies that can potentially be used clinically to alleviate the neuropsychiatric symptoms and cognitive decline in AD course.

## 1. Introduction

Alzheimer’s disease (AD) is a progressive neurodegenerative disorder that is characterized by the pathological accumulation of extracellular β-amyloid (Aβ) plaques and intracellular neurofibrillary tangles, leading to severe brain atrophy and irreversible cognitive decline [1]. Currently, restoring the cholinergic system with cholinesterase inhibitors is the primary therapeutic option for improving cognition in mild and moderate AD [2]. However, in addition to cognitive impairment, an estimated 12.8–66.0% of mild cognitive impairment (MCI) patients have various types of behavioral and psychological symptoms of dementia (BPSD) [3], and approximately 90% of AD patients suffer from BPSD throughout the disease course, including emotional (e.g., depression, anxiety, euphoria, and apathy), verbal, motor, and vegetative symptoms [4]. Unfortunately, immunotherapy against Aβ or cholinergic therapy has shown limited effects in alleviating neuropsychiatric symptoms (NPS) in several clinical trials for AD patients [5,6]. Thus, it is urgent to understand the initiation and progression of NPS in AD courses and to develop effective novel treatments for AD psychosis.

The involvement of serotonin (5-hydroxytryptamine, 5-HT) in the regulation of emotion and cognition has been well documented in the growing body of literature [7,8,9]. 5-HT is originally synthesized in midbrain raphe neurons expressing tryptophan 5-hydroxylase 2 (TPH2) and, subsequently, degraded to 5-hydroxy-3-indoleacetic acid (5-HIAA) by monoamine oxidase A [10]. Anatomically, the midbrain serotonergic neurons are organized into nine distinct clusters, with the dorsal raphe nucleus (DRN) and median raphe nucleus (MRN) representing the two largest neuronal populations [10]. 5-HT neurons within the DRN and MRN exhibit significant heterogeneity in multiple aspects, including cellular morphology, projection patterns, and electrophysiological and neurochemical properties [10,11,12]. In addition, 5-HT can bind to different postsynaptic 5-HT receptors (5-HTRs), including 14 subclasses from 7 families [13]. Overall, the intricate structural organization of the serotonergic system confers both functional diversity and selective vulnerability to pathogenic stimuli upon 5-HT neurons. Notably, serotonin metabolism and neurotransmission are profoundly altered in the hippocampus and cortex of AD patients and mouse models, which may contribute to the emotional and cognitive deficits in AD [14]. Selective serotonin reuptake inhibitors (SSRIs) not only prevent the accumulation of Aβ and Tau [15,16], but also reverse the emotional and cognitive deficits in AD patients and mouse models [17,18,19], indicating that targeting the serotonergic system could be an effective approach to treat cognitive and psychiatric symptoms of AD.

In addition, 5-HT neurons make up only a small proportion of the neuronal architecture in the DRN and MRN [10], and the ascending projections from raphe nuclei to the forebrain are predominantly glutamatergic neurons, the majority of which express vesicular glutamate transporter 3 (vGluT3) [20]. Accumulating evidence suggests that vGluT3-positive raphe neurons are involved in the regulation of serotonergic transmission and hippocampal synaptic plasticity [21,22]. Particularly, strengthening the raphe–hippocampal glutamatergic circuit enhances memory retrieval in a mouse model of AD [23]. Thus, apart from the serotonergic circuit, glutamatergic projections from raphe also play a pivotal role in regulating hippocampal activity, but current knowledge on the role of disrupted raphe–hippocampal glutamatergic circuits in AD is lacking. In this review, we focus on the role of the raphe–hippocampal serotonergic and glutamatergic circuits and their deficits in AD. We first summarize the diversity of the raphe nucleus and the architecture of raphe–hippocampal circuits. We then elucidate the role of raphe–hippocampal serotonergic and glutamatergic circuits in regulating hippocampal activity, as well as emotional and cognitive behavior. Additionally, we explore the role of aberrant raphe–hippocampal circuitry in AD-associated emotional and cognitive deficits, highlighting that stimulation of the raphe–hippocampal circuit could be a valuable strategy to prevent the development of AD.

## 2. Anatomical, Neurochemical, and Electrophysiological Diversity of Raphe Nuclei

Raphe nuclei, characterized by 5-HT neurons, are evolutionarily conserved brainstem nuclei that exist in both invertebrates and vertebrates. Historically, the first description of raphe nucleus was made by Santiago Ramón y Cajal, with a drawing showing the cell morphology in the subaqueductal nucleus of the raphe [14]. However, the cellular profile in the raphe nucleus remained unclear until Dahlstöm and Fuxe demonstrated the existence of monoamine-containing neurons in the central nervous system, and they subsequently classified the 5-HT-immunopositive neurons into nine clusters (B1–B9) [24]. These clusters were further subdivided into two groups by Jacobs and Azmita [25]. Namely, the superior group, including the DRN (B6–7), the caudal MRN (B5), the caudal liner nucleus, the rostral MRN, the prepontine raphe nucleus (B8), and the supragenual raphe nucleus (B4), which send projection fibers to the brainstem and the forebrain, while neurons in the inferior group, including nucleus raphe pallidus (RPa, B1), nucleus raphe obscurus (ROb, B2), ventrolateral medulla, and nucleus raphe magnus (RMg, B3), project to the cerebellum and spinal cord (Figure 1) [14,25]. The anatomical classification of the raphe nucleus reflects the topographical organization of raphe neurons. It is important to note that these distinct clusters imply cellular or functional differences between the raphe neurons.

Investigations into the serotonergic system have disclosed the neurochemical, hodological, and electrophysiological diversity of raphe neurons [10]. The majority of 5-HT neurons are located in the DRN and MRN. Anterograde tracing using P. vulgaris leucoagglutinin (PHA-L) has shown that 5-HT neurons in the DRN and MRN differ significantly in their axonal morphology and projection pattern [11,26]. MRN-derived serotonergic neurons send irregularly spaced thick fibers with large and spherical varicosities (beaded fibers, BFs) [11,14,27], which mainly innervate midline brain regions, including dorsal and ventral hippocampus (vHP) [28,29], medial septum (MS) [30], and lateral hypothalamus [31]. In contrast, DRN-derived serotonergic axons are regularly spaced fine fibers (FFs) with typically small pleomorphic varicosities that are granular or fusiform in shape (Figure 2) [11,14]. FFs from DRN are uniquely connected to lateral brain regions associated with emotion and motivation (e.g., amygdala, nucleus accumbens, ventral tegmental area, ventral pallidum, and vHP) [31,32,33,34,35]. Interestingly, the caudal third of the DRN (hereafter referred to as the B6 region) shares similar connectivity with the MRN (B8); both of which dominate over the septum and hippocampus [31]. Thus, DRN and MRN output fibers innervate forebrain structures in a non-overlapping manner.

Meanwhile, DRN and MRN have a similar input pattern but display subtle differences in connection strength. For example, MRN^5-HT^ neurons receive fewer afferents from the prefrontal cortex, with a preference for inputs from midline areas such as MS, hypothalamus, and brainstem [37,38,39,40]. Taken together, the efferent and afferent fibers of MRN correspond to midline structures, whereas those of DRN correspond to lateral structures (Figure 3). The different topographic organization of DRN and MRN circuits probably underlies the different neuronal properties and behavioral features.

In fact, 5-HT neurons represent only a small fraction of raphe nuclei [41,42]. Immunohistochemical studies have discerned that 5-HT neurons make up 25–30% of the total neuronal population in the DRN [37,43]. High-throughput single-cell transcriptome profiling of the DRN has uncovered five neuronal clusters in decreasing order of abundance: serotonergic, dopaminergic, GABAergic, glutamatergic, and peptidergic neurons [44]. Several molecularly defined subtypes of 5-HT neurons are found to co-express glutamic acid decarboxylase (GAD) or vGluT3, indicating that these 5-HT subpopulations co-release serotonin, gamma-aminobutyric acid (GABA), or glutamate [45]. Notably, 5-HT neurons only account for a minority of the total neuronal population (8.5%) in the MRN, while 26% of neurons are vGluT3, 12.8% are 5-HT and vGluT3 co-expressing neurons, 37.2% of neurons are GABAergic, and 14.4% are triple-negative [41]. Anterograde and retrograde tracing showed that ascending projections from MRN to the forebrain were predominantly vGluT3 neurons [20,46], reinforcing the notion that vGluT3 neurons make up the majority of projection neurons in the MRN [47].

Spatial single-cell RNA sequencing has identified gene expression in the subdomains of the DRN, such as lateral DRN neurons projecting to the colliculus and auditory brainstem nuclei, which express *Pet1*, *Tph2*, *Sert*, and *En1*. Dorsolateral DRN neurons express *Pet1*, *Tph2*, *Sert*, *Drd2*, *Gad2*, *Pax5*, and *En1*. Dorsal DRN neurons projecting to the central amygdala and the paraventricular nucleus of the hypothalamus express *Pet1*, *Tph2*, *Sert*, and *En1*. Ventromedial DRN neurons projecting to the frontal cortex express *Pet1*, *Tph2*, *Sert*, *vGluT3*, and *En1*. Cadual DRN neurons projecting to the septum and vHP express *Pet1*, *Tph2*, *Sert, Met, and En1* [10]. Apart from the same gene expression of *Pet1*, *Tph2*, and *Sert*, MRN neurons projecting to the hippocampus, septum, and suprachiasmatic nucleus express Hoxa2, Met, Oxtr, Nr2f2, vGluT3, and Tacr3, while MRN neurons projecting to the dorsal tegmental nucleus of Gudden express *Nmbr* and *Chrna4* [10], suggesting that 5-HT neurons exhibit molecular diversity. It is, therefore, convenient for researchers to label and modulate serotonergic neurons with genetically modified mouse models, such as Pet1-Cre [48,49], Tph2-Cre [50], Sert-Cre [51], and 5-HT_1A_-iCre mice [52].

The intrinsic electrical properties of 5-HT neurons depend on factors such as developmental stage, anatomical location, gene expression, and projection pattern. First, the electrical properties of 5-HT neurons change significantly during the first 3 weeks of life [53]. Due to a lack of 5-HT_1A_, autoreceptors, and GABA input, 5-HT neurons are hyperexcitable at postnatal day 4 (P4), 5-HT_1A_R-mediated auto-inhibition starts to develop by P21 when 5-HT neurons give rise to an increase in spontaneous spikes [53]. Second, 5-HT neuron activity is associated with anatomical location, such as the DRN versus the MRN [12], and the lateral DRN versus the medial DRN [54]. In vitro patch-clamp electrophysiological studies have demonstrated that 5-HT neurons in the lateral wing of the DRN are more excitable than ventromedial DRN^5-HT^ neurons [54]. It is important to note that the difference in 5-HT neuron activity between the DRN subregions may be relevant to the input strength of presynaptic partners [39]. Moreover, the electrical properties of DRN^5-HT^ neurons are largely different from those of the MRN, with MRN^5-HT^ neurons having a larger afterhyperpolarization amplitude and a shorter time constant than those of the DRN [12]. Mechanistically, the inhibitory effects elicited by the activation of 5-HT_1A_ autoreceptors are stronger in DRN^5-HT^ neurons than in the MRN; furthermore, non-serotonergic neurons are responsive to 5-HT_1A_R agonists in the DRN but not in MRN non-serotonergic neurons [12].

DRN and MRN neurons are genetically divided into several clusters, and the electrical properties of raphe neurons vary among these clusters. In the MRN, Hoxa2-Pet1 neurons are more excitable than En1-Pet1 and Egr2-Pet1 neurons [55], and vGluT3 neurons fire faster and more variably than 5-HT neurons [56]. Moreover, 5-HT neurons projecting to the basolateral amygdala fired more frequently than those projecting to the medial prefrontal cortex (mPFC) and the dorsal hippocampus, suggesting that the intrinsic electrical properties of 5-HT neurons are linked to efferent projection target [26]. The electrophysiological diversity of 5-HT neurons indicates that the 5-HT neurons respond differently to external stimuli.

## 3. Architecture of Raphe Nucleus–Hippocampal Circuits

Ascending fibers from the DRN and MRN rarely innervate the same brain structure, with the exception of MS and hippocampus. Previous studies have shown that both DRN and MRN send dense serotonergic and non-serotonergic afferents to the hippocampus [57,58], but these ascending fibers to the hippocampus differ between subdivisions of DRN and MRN. Retrograde tracing combined with 5-HT immunohistochemistry has shown that a small number of retrogradely labeled 5-HT neurons are found in the B6 region, and a large number of 5-HT neurons are found in the B8 region [57]. Anterograde tracing of individual groups (B5–B9) of raphe nuclei confirmed the similar projection pattern between the B6 region of DRN and MRN [59], but the most notable differences were observed in the projections from the caudal DRN and the rostral DRN, with moderately dense afferents from the caudal DRN to hippocampus and none from the rostral DRN [60]. Interestingly, the forebrain afferents to the raphe nuclei showed disparity between rostral 2/3 of the DRN and the caudal third of the DRN, but similar afferent innervation patterns between B6 region of DRN and MRN [61]. Thus, the anterograde and retrograde tracing together suggest that the B6 region shares the similar afferent and efferent patterns with the MRN. This argument is supported by the same developmental origin that the B6 region and MRN are defined by the common expression of Engrailed-1 (En1, also called rhombomere 1) [62,63], as well as the same genetic lineage that Met is expressed in both the B6 region and MRN [55]. On the other hand, anatomically, the B6 region and MRN are not connected in the coronal view; however, they are closely connected in the sagittal section (Figure 1) [31].

The B6 region, in striking contrast to the rostral portion of DRN, sends projections to the dorsal and vHP [59,60]. It is assumed that the B6 region preferentially innervates the vHP. In support of this assumption, microdialysis experiments showed that electrical stimulations of the DRN (5 Hz, 300 μA, 20 min) evoked a short-lasting 5-HT release (an increase by 70–100%) in the vHP, but the dorsal hippocampus did not respond to the DRN stimulation, whereas stimulation of the MRN induced 70–100% of the 5-HT release in the dorsal hippocampus, vHP, and MS [29]. The assumption is further reinforced by the experiment that optogenetic activation of DRN^5-HT^ neurons caused brain-wide activation, including the mPFC, striatum, and ventral tegmental area, but excluding the dorsal hippocampus [64]. In fact, DRN^5-HT^ neurons are also involved in the regulation of hippocampal activity. For example, selective activation of serotonergic neurons in the DRN by infusion of substance P increased extracellular levels of 5-HT by 30% for 20 min in the vHP compared to the vehicle control [65], whereas perfusion of tetrodotoxin into the DRN simultaneously decreased hippocampal 5-HT release [66]. Chemogenetic activation of DRN neurons projecting to the vHP or axonal terminals in the vHP reduced c-Fos (a marker of neuronal activity) expression in the vHP via activation of postsynaptic 5-HT_1B_R [35]. In addition, DRN^5-HT^ neurons preferentially reactivate neuronal ensembles in the dorsal dentate gyrus (DG) by activating dopaminergic signaling in the ventral tegmental area [67]. DRN-derived serotonergic fibers also project to the dorsal hippocampal CA1 and modulate excitability of CaMKⅡ-positive neuron via 5-HT_1B_R or 5-HT_4_R [68]. DRN^5-HT^ neurons make synaptic contact with 5-HT_3A_R-positive interneurons, and dysfunction of DRN^5-HT^-HP^HTR3A^ transmission results in an anxiety phenotype [69].

Among the raphe nuclei, MRN neurons establish strong synaptic contact with the hippocampal region. Retrograde tracing combined with 5-HT immunohistochemistry has shown that only half of the retrogradely labeled MRN neurons projecting to the hippocampus belong to 5-HT neurons, whereas the majority of retrogradely labeled neurons are vGluT3^+^5-HT^−^ neurons [70]. We and others have unveiled that serotonergic fibers from the MRN are largely distributed across the border between the stratum lacunosum moleculare (SLM) and stratum radiatum (SR) of the CA1, as well as in the subgranular zone in the hilus [49,58]. Both serotonergic and non-serotonergic afferents from the MRN have been shown to climb along proximal dendrites and soma of calbindin-positive interneurons [58,71,72], suggesting that GABAergic interneurons are the major postsynaptic target of MRN neurons [71]. For example, MRN^vGluT3^ neurons make direct synaptic contact with parvalbumin (PV) interneurons, thereby controlling the efficacy of spatial memory retrieval [23]. Both MRN^vGluT3^ and MRN^5-HT^ neurons can activate 5-HT_3A_R-positive interneurons, thereby inducing a long-lasting increase in GABAergic transmission in CA1 pyramidal neurons [22,49]. MRN^vGluT2^ neurons selectively innervate septal PV interneurons that project to hippocampus; therefore, in vivo stimulation of MRN^vGluT2^ neurons instantly promotes memory acquisition in mice [7]. Another cluster of GABAergic interneurons, neuropeptide Y (NPY)-expressing interneurons, receives a serotonergic input from the MRN [73] and indirectly controls the firing of the hippocampal output neurons. Additionally, PV interneurons were selectively innervated by septal GABAergic neurons but avoided by MRN^5-HT^ neurons [74], whereas a small number of cholecystokinin (CCK) interneurons receive input from both MRN and septal afferents [72]. In the raphe–hippocampal circuits, single or double synaptic contacts were more common on vasoactive intestinal polypeptide (VIP)-positive interneurons, which may ensure the rhythmic synchronization of inhibitory tones in the hippocampus [28].

## 4. Synaptic Control of Hippocampal Activity by the Raphe Nucleus

### 4.1. Fast Modulation of Hippocampal Inhibitory Network by Raphe Nucleus

The aforementioned architecture of raphe–hippocampal circuits can be summarized by the synaptic connection between MRN and hippocampal GABAergic interneurons, including calbindin-containing, PV, CCK, VIP, and NPY, as well as 5-HT_3A_R-expressing interneurons, which underly the fast synaptic control of the hippocampal inhibitory network. Therefore, it is the case that hippocampal inhibitory network is under control of the raphe nucleus. First, serotonin can attenuate feedback inhibition onto interneurons [75]. Increased GABA levels were found in the hippocampus of TPH2 knockout mice, indicating that 5-HT inhibits hippocampal GABA release [76]. Subsequently, Kao et al. found that electrical stimulation of MRN inhibited the majority (60%) of the GABAergic interneuron activity in the CA1 region [77]. These GABAergic interneurons located in the oriens lacunosum-moleculare (hereafter referred to as O-LM interneurons) of the hippocampal CA1 region can be classified into a caudal ganglionic eminence-derived subpopulation expressing 5-HT_3A_R and a medial ganglionic eminence-derived subpopulation lacking 5-HT_3A_R [78]. 5-HT released from serotonergic terminals profoundly and reversibly reduces the excitatory input from local CA1 collaterals to 5-HT_3A_R-expressing O-LM interneurons, and this effect can be mimicked by fenfluramine [75]. Mechanistically, 5-HT, via activation of presynaptic 5-HT_1B_R, substantially inhibits the excitation of CCK interneurons in the CA1 area, thereby selectively reducing feedback inhibition onto pyramidal neurons and allowing for an increase in the integration of time windows for spike generation by CA1 pyramidal neurons [79]. Overall, subcortical serotonergic neurons can control the hippocampal excitatory output by modulating inhibitory strength.

The innervation of hippocampus by serotonergic neurons depends on multiple postsynaptic 5-HTRs. 5-HT excites GABAergic interneurons through activation of ionotropic 5-HT_3_Rs [80]. Varga et al. have shown that selective stimulation of MRN^5-HT^ neurons produced rapid activation of hippocampal interneurons, and MDL72222 (a 5-HT_3A_R antagonist) only reduced the amplitude of excitatory input into interneurons, whereas a combination of both 5-HT_3A_R antagonists and glutamate receptor blockers can fully abolish the excitation of hippocampal interneurons, suggesting that serotonergic neurons co-release 5-HT and glutamate to excite GABAergic interneurons [81]. In actuality, MRN^vGluT3^ neurons send primary afferents to the hippocampus. Likewise, MRN^vGluT3^ neurons can activate 5-HT_3A_R-expressing GABAergic neurons at the SR/SLM border, and this process is negatively regulated by 5-HT/5-HT_1B_R signaling [22]. In addition, 5-HT_2A_Rs are predominantly, but not solely, present in the interneurons located at the border of SR/SLM of the CA1 region [82], and 5-HT can stimulate GABA release via activation of 5-HT_2A_R signaling [83]. Serotonergic neurons have also been found to establish multiple synaptic contacts with NPY interneurons that innervate the distal dendrites of principal cells, thereby controlling the excitability of hippocampal output neurons [73]. Meanwhile, serotonergic neurons also innervate VIP interneurons that mediate feedforward inhibition onto GABAergic interneurons and control rhythmic synchronization of inhibitory networks [28]. Altogether, whether the profound effect of raphe–hippocampal pathway is mediated by GABAergic interneurons or via a direct action on excitatory neurons remains to be determined [74], but it is clear that the serotonergic and glutamatergic input from MRN governs rapid modulation of the hippocampal inhibitory network.

### 4.2. Subcortical Control of Hippocampal Theta-Rhythm by Raphe Nucleus

Theta rhythm is an oscillatory neural activity that fires at a frequency of 3–10 Hz in the hippocampus and is deeply implicated in mnemonic, motor, and sensory functions [84,85,86]. A growing body of evidence suggests that activation of the MRN suppresses theta rhythms, whereas inhibition of the MRN induces persistent theta [87,88]. Interestingly, there is a dissociable pathway to facilitate theta and non-theta states in the raphe–hippocampal circuit. Selective inactivation of the MRN serotonergic output with 5-HT_1A_R agonist (8-OH-DAPT) induces hippocampal theta rhythm, whereas activation of non-serotonergic (possibly glutamatergic) MRN-hippocampal pathway promotes hippocampal theta generation [87]. During hippocampal theta oscillations, MRN^5-HT^ neurons exhibit little or no theta-related activity, whereas most GABAergic neurons and vGluT3 neurons in the MRN show increased and decreased activity, respectively [89,90]. GABAergic neurons establish a monosynaptic connection with local serotonergic and vGluT3 neurons; therefore, even if MRN GABAergic neurons do not directly innervate the hippocampus, they could still modulate hippocampal theta oscillations by regulating the activity of local vGluT3 and serotonergic neurons [89,91]. Pharmacological evidence has shown that infusion of an N-methyl-D-aspartate (NMDA) receptor antagonist (MK-801) into the MRN or a GABA_A_ receptor agonist (muscimol) facilitates the hippocampal theta oscillation in urethane-anesthetized rats [88,92,93]. Collectively, these results suggest that the local inhibitory circuit in the MRN may synchronize ascending serotonergic/glutamatergic modulation with hippocampal theta rhythm on a sub-second timescale [94].

Hippocampal theta rhythm is modulated by synchronizing afferents from the MS and desynchronizing afferents from the MRN. Inhibition of the MS suppresses hippocampal theta rhythm, whereas inhibition of the MRN produces sustained theta, and this effect can be reversed by inhibiting the MS, suggesting that MRN–MS–hippocampal circuit promotes different brain states (theta or non-theta) [87]. 5-HT depletion in the hippocampus resulted in an increased expression of high-frequency theta activity, which facilitates spatial learning in the Morris water maze [95]. Concomitantly, 5-HT depletion in the MS induced both low-frequency (4.5–6.5 Hz) and high-frequency (6.5–9.5 Hz) theta activity [95]. Additionally, 5-HT depletion in the supramammillary or posterior hypothalamic nucleus results in theta rhythm alterations in the CA1 region [96]; infusion of 8-OH-DAPT into the pedunculopontine tegmental nucleus induces prolonged spontaneous theta rhythm [97]. These results suggest that ascending serotonergic fibers indirectly modulate hippocampal theta rhythm through synaptic contacts with other nuclei, such as MS, supramammillary nucleus, and posterior hypothalamic nucleus.

### 4.3. Subcortical Control of Synaptic Plasticity in the Hippocampus

Theta rhythm has been shown to regulate long-term potentiation (LTP) in the hippocampus [98], strongly suggesting that ascending projections from the raphe nucleus may also control hippocampal synaptic transmission and plasticity. Likewise, the effects on hippocampal LTP induction vary across different neuronal types of the MRN. Optogenetic activation of serotonergic terminals in the CA1 region enhances excitatory transmission in both Schaffer collateral pathway and LPP-DG pathway [99,100]. In contrast, activation of MRN^vGluT3^ afferents suppresses LTP induction in the CA3–CA1 pathways in male mice [22]. The underlying molecular mechanism is also different, with the serotonergic enhancement of LTP acting through activation of 5-HT_4_R, whereas glutamatergic suppression of LTP occurs through activation of GABAergic interneuron-mediated feedforward inhibition [22,99].

## 5. Raphe–Hippocampal Serotonergic and Glutamatergic Circuits Regulate Emotional Behavior

### 5.1. Anxiety

It is well established that DRN^5-HT^ neurons play a pivotal role in regulating anxiety behavior through their projections to emotion-related nuclei, such as amygdala [32,101], bed nucleus of the stria terminals [102], and vHP [69]. Although the role of the MRN in the regulation of anxiety has received less attention than that of the DRN, there is increasing evidence that MRN–hippocampal serotonergic and glutamatergic circuits control anxiety behavior. Previous studies have shown that electrolytic or pharmacological lesions of MRN exert an anxiolytic effect in both restraint and nonrestraint rats [103]. Conversely, Netto et al. showed that lesions of MRN^5-HT^ neurons produced anxiogenic behavior in chronically restrained rats [104]. Although the role of MRN in the regulation of anxiety remains controversial, emerging evidence supports the assumption that activation of MRN^5-HT^ neurons promotes anxiety. For example, selective optogenetic activation of MRN^5-HT^ neurons enhances anxiety-like behavior [105]; inhibition of MRN^5-HT^ neurons promotes anxiolysis [106], whereas activation of DRN^5-HT^ neurons has antidepressant-like effects [105,107]. Activation or inhibition of MRN^5-HT^ neurons via infusion of 5-HT_1A_R agonists/antagonists yielded similar results, with 8-OH-DAPT inducing anxiolytic effects, whereas WAY-100635 induced the opposite effects [108,109,110]. Another critical question is: How does the MRN modulate anxiety behavior? Electrophysiological recordings combined with in vivo microdialysis confirmed that pharmacological or optogenetic stimulation of MRN^5-HT^ neurons produced rapid 5-HT release in the dorsal hippocampus, which may enhance anxiety [105,110,111]. In support of this view, simultaneous injections of 5-HT-acting drugs into the MRN and dorsal hippocampus demonstrated that the MRN regulates anxiety via serotonergic innervation of the dorsal hippocampus [106].

Ascending glutamatergic afferents from raphe to the hippocampus are also involved in the modulation of anxiety. DRN and MRN contain a mass of vGluT3 neurons projecting to the hippocampus [46]. vGluT3 knockout mice showed increased anxiety behavior [21,112], whereas activation of MRN^vGluT3^ neurons caused anxiolytic effects [112]. Notably, most of vGluT3 immunopositivity is found on nerve terminals in the hippocampus [113], and the vGluT3 positively modulates 5-HT transmission by decreasing 5-HT_1A_R-mediated autoinhibition [21]. Therefore, deletion of vGluT3 is likely to excite 5-HT neurons and thereby promote anxious behavior. In contrast, DRN vGluT3 and 5-HT double-positive neurons control separate behavioral features of anxiety via their projection to the amygdala but not the hippocampus [32]. Thus, DRN^vGluT3^ and MRN^vGluT3^ neurons modulate anxiety via different pathways.

### 5.2. Depression

Hypoactive 5-HT neurons have been implicated in the pathophysiology of depressive-like behaviors [114]. However, recent studies have suggested that MRN and DRN neurons respond differently to depression symptoms. A hyperactivated DRN with overactive projections to the amygdala, striatum, and periaqueductal gray, and a hypoactive MRN with underactive projections to the hippocampus have been found in patients with major depressive disorder (MDD) [115]. Normally, MRN input to the hippocampus inhibits aversive memory consolidation, and increased hippocampal hyperactivity in MDD may result from the disrupted MRN–hippocampal pathway [115]. It has been suggested that an activated MRN^5-HT^-DG^CamKII^ pathway underlies behaviors of despair in mice subjected to a regimen of chronic behavioral despair [116]. MRN contains a large population of vGluT2-expressing neurons that receive input from the negative experience-related nucleus and project to the aversion centers, and these neurons were selectively activated by aversive stimuli, thereby inducing depression-related anhedonia [7]. Overall, MRN serotonergic and glutamatergic afferents to the hippocampus may promote depression. Therefore, it is the case that simulation of the MRN decreased locomotion and increased depressive-like behaviors [117].

In contrast to the MRN, activation of DRN^5-HT^ neurons has antidepressant-like effects [107]. DRN^5-HT^ can retrieve positive memory ensembles in the DG mediated by 5-HT-activated dopaminergic signaling from the ventral tegmental area, providing insight into how DRN^5-HT^ neurons elicit antidepressant effects [67]. In addition, reduced serotonergic transmission from the DRN to LHb neurons would increase LHb activity and contribute to depressive symptoms; optogenetic activation of the DRN–LHb circuit mitigates the depressive symptoms in a rat model of chronic unpredictable mild stress [118].

### 5.3. Aggression

In the literature, 5-HT is thought to play an inhibitory role in controlling aggression. Inhibition of 5-HT synthesis by genetic ablation of TPH2 reduces anxiety but exaggerates aggression in male rats [119,120]. Similarly, selective lesions of DRN^5-HT^ or MRN^5-HT^ neurons promote submissive cats to engage in dominance fights, and biochemical analysis revealed a significant reduction in 5-HT and 5-HIAA in the hypothalamus, the amygdala, and the hippocampus in either MRN- or DRN-lesioned rats [121]. Furthermore, 5-HT levels were decreased in the prefrontal cortex, the striatum, the amygdala, the hippocampus, and the DRN in stressed animals compared to non-stressed animals [120]. This evidence was sufficient to link the reduced cerebral 5-HT levels and serotonergic activity to exaggerated aggression. Given that chronic social defeat paradigm simultaneously induced a marked increase in serotonin transporter (SERT) expression in the DRN and CA1/CA3 of the hippocampus [122], raising the possibility that the raphe–hippocampal pathway engages in the regulation of aggression. Indeed, while the hyperactivated raphe–hippocampal pathway is involved in the stress-evoked escalated aggression, chemogenetic activation of the DRN–vHP pathway reduces reactive aggression via suppression of vHP activity by activating 5-HT_1B_R, then the vHP suppresses the downstream ventromedial hypothalamus (VMH), which is the core aggression center [35]. Infusion of 5-HT_1B_R agonists (anpirtoline) into the vHP reduces neuronal activity projecting to VMH and reactive aggression [35]. These results suggest that DRN–vHP inhibits aggression via innervation of VMH.

### 5.4. Addiction and Reward

The serotonergic system dynamically interacts with the hippocampus to regulate reward-related information processing. DRN^5-HT^ neurons are responsible for encoding the reward signal by co-releasing 5-HT and glutamate [9,123,124,125]; these neurons are selectively recruited by emotional stimuli rather than neutral stimuli, and subsequently, they activate the downstream ventral tegmental area but not the hippocampus [126]. Meanwhile, in vivo two-photon imaging has screened two distinct neuronal populations in the MRN that project to the hippocampus: one is associated with reward delivery and the other with locomotion, and stimulation of these hippocampal projection fibers modulates reward-induced behavior [127]. During successful goal-directed behavior, there is an increase in serotonergic transmission from MRN to the vHP, which suppresses vHP activity by activating 5-HT_3A_R. It can be concluded that suppression of the vHP via MRN^5-HT^-vHP^5-HT3AR^ is required for successful goal-directed behavior [128]. Overall, these findings suggest that DRN and MRN target different nuclei to modulate reward signaling.

Alcohol addiction is associated with an alteration in 5-HT neuronal function, with a transition from the DRN-dependent regulation of short-term alcohol intake to the MRN-dependent regulation of long-term alcohol intake [129]. In mice with long-term voluntary alcohol consumption, the serotonergic fibers projecting from MRN to the DG were severely affected by the altered expression of 5-HT_1A_ autoreceptors in MRN^5-HT^ neurons [129]. Chemogenetic inhibition of hippocampal serotonergic fibers from MRN^5-HT^ neurons alleviates alcohol addiction after long-term alcohol consumption [129].

## 6. Subcortical Modulation of Memory by Raphe–Hippocampal Circuits

### 6.1. Aversive Memory

Repeated aversive stimuli, such as enemy odor, foot electric shock, and pain, help to form and consolidate aversive (fear) memories. During this process, serotonergic and glutamatergic afferents from the MRN–hippocampus are motivated to calibrate the memory storage. MRN is believed to control the fear memory via innervation of amygdala, hippocampus, and prefrontal cortex. Either acute or chronic lesions of the MRN clearly suppressed freezing behavior in rats [130]. In contrast, electrical or optical stimulation of the MRN alone consolidates remote but not recent fear memory traces, and optogenetic silencing of the MRN during foot shock, which attenuates conditioned fear memories [131]. Next, it must be asked whether the serotonergic projections from MRN–HP influence fear memories. The results show that infusion of the 5-HT_1A_R agonist or the corticotropin-releasing factor receptor antagonist into either MRN or hippocampus abolished contextual freezing, suggesting that ascending 5-HT fibers from MRN to the hippocampus are implicated in the acquisition of fear memories [132,133]. It has been proposed that stimulation of the MRN interferes with fear memory consolidation by suppressing hippocampal sharp-wave ripple oscillations [134]. Meanwhile, suppression of MRN^vGluT2^ neurons impaired both contextual and cued fear memory formation via activation of septal–hippocampal inhibitory circuits [134]. Furthermore, activation of MRN^vGluT3^ neurons enhances the GABAergic input to pyramidal neurons in the mPFC and selectively attenuates fear memories in female but not male mice [135]. Therefore, it can be concluded that innervation of the hippocampus by MRN provides bottom-up control of fear circuits, complementing the top-down control by mPFC [131,136,137].

### 6.2. Spatial Memory

The serotonergic system plays a crucial role in the regulation of spatial memory through its projections to the hippocampus. Acute depletion of tryptophan, the primary amino acid used to synthesize 5-HT, impairs memory consolidation but improves focused attention in healthy individuals [138,139]. Postnatal depletion of 5-HT via administration of parachlorophenylalanine (PCPA, an inhibitor of 5-HT synthesis) not only causes neuronal loss in the hippocampus but also impairs spatial memory during adulthood [140]. In line with this finding, global depletion of cerebral 5-HT (Pet1 knockout mice) leads to significant deficits in novel object recognition memory [141]. Taken together, these findings underscore that 5-HT is essential for memory consolidation.

The literature indicates that 5-HT can modulate hippocampal activity in a bidirectional manner, either through direct synaptic contact with hippocampal neurons or indirectly via MS cholinergic/GABAergic inputs to the hippocampus. Selective depletion of 5-HT in MS disrupts the septal-hippocampal information relay [95,142]. Infusion of 5, 7-dihydroxytrptamine (5, 7-DHT) into the MS potentiates the acquisition of spatial information and working memory [95,142]. Additionally, hippocampal 5-HT depletion facilitates place learning in the Morris water maze [95]. However, simultaneous lesions of septal cholinergic innervation of the hippocampus and the serotonergic pathway result in a reduction in hippocampal inhibition and behavioral deficits, and intrahippocampal raphe grafts can ameliorate the spatial memory deficits in rats combined with serotonergic and cholinergic lesions [143,144], suggesting a complementary role for the serotonergic innervation of the hippocampus after loss of septal cholinergic innervation of the hippocampus.

Stimulation of MRN^5-HT^ or DRN^5-HT^ neurons produces conflicting results in spatial memory impairment. It has been reported that inhibition of serotonergic neurons from the MRN, but not DRN, is sufficient to impair object recognition in adult mice [141]. Electrolytic lesions of the MRN have a profound effect on both acquisition and retention in the eight-arm maze task [145]. Moreover, optogenetic activation of serotonergic terminals in the CA1 region enhances excitatory transmission at the CA3–CA1 pathways and spatial memory in mice, whereas optogenetic silencing of serotonergic terminals impairs spatial memory [99]. Blockade of hippocampal activity via activation of 5-HT_1A_ heteroreceptors impairs the acquisition and performance of a spatial task in the water maze [146]. In contrast, several studies have found that inactivation of either the MRN or the hippocampus has no effect on spatial memory retrieval and consolidation [147,148]. Whether serotonergic innervation of the hippocampus is necessary for regulating hippocampal synaptic plasticity and spatial memory remains to be further determined [141]. The different results may depend on MRN cell types, animal models, stimulation methods, etc. For example, it has been found that stimulation of MRN^vGluT3^ neurons improves spatial memory retrieval by strengthening MRN^vGluT3^-HP^PV^ synaptic transmission in AD mice [23].

Interestingly, inactivation of serotonergic neurons from the DRN attenuates the spatial learning deficits in mice with the loss of cholinergic innervation of hippocampus [149,150], whereas blocking orexin 1 receptors in the DRN impairs spatial memory consolidation in rats [151]. These results suggest that DRN^5-HT^ neurons also participate in regulating spatial memory but in a manner opposite to that of MRN neurons.

### 6.3. Social Memory

In the hippocampus, CA2 pyramidal neurons act as a hub receiving a variety of inputs from subcortical areas, including MS, supramammillary nucleus, and MRN, thereby modulating social aggression and memory [152]. However, the serotonergic circuit between raphe and hippocampus is remodeled under social stress. Chronic social stress markedly reduced 5-HT neuronal activity and hippocampal 5-HT reuptake [153,154]. In contrast, stimulation of the MRN increased friendly social interactions in the resident intruder test [117]. 5-HT released from the MRN not only promotes social interaction but also consolidates social memory. Stimulation of MRN^5-HT^ neurons bidirectionally regulates social memory by activating dorsal CA2-projecting MS glutamatergic neurons [155]. With the exception of serotonergic neurons, both GABAergic and dopaminergic neurons in the MRN contribute to social interactions in mice but have no effect on social memory [156,157].

## 7. Abnormal Raphe–Hippocampal Circuits in AD Patients and Mouse Models

AD is the most common type of neurodegenerative diseases in the elderly, and it has been well established that the clinical symptoms of AD are characterized by β-amyloid (Aβ) plaques, phosphorylated Tau, and gliosis in the cerebral parenchyma, resulting in progressive cognition decline [158]. In addition to the obvious cognitive deficits, AD patients also suffer from anxiety [159], depression [160] and sleep disturbances [161]. These psychiatric disorders have linked the disrupted raphe–hippocampal circuitry to cognitive decline in AD (Figure 4).

### 7.1. The 5-HT Neurons in AD Patients and Mouse Models

Approximately 90% of AD patients suffer from NPS years prior to cognitive decline, and recent evidence has linked the early behavioral symptoms to pathological changes in the raphe nucleus [3]. The loss of 5-HT neurons in the raphe nuclei, including raphe obscurus and pallidus, MRN, and DRN, has been reported in the early stages of AD (Table 1) [162,163]. Immunostaining for TPH2 showed that although no significant difference was found between young and aged dogs without cortical Aβ deposits, aged dogs with cortical Aβ pathology had 33% fewer 5-HT neurons in the DRN and MRN than those without cortical Aβ pathology [164]. However, in AD mouse models of overexpressing human amyloid precursor protein (hAPP), the results from our group and others have shown that there was no significant difference in the number of 5-HT neurons in either DRN or MRN between non-transgenic and hAPP mice [49,165]. It can be concluded that aging is not responsible for the degeneration of 5-HT neurons, but that the degeneration of serotonergic system correlates with the development of cortical Aβ deposits.

Pathohistological studies have shown that the early onset of tauopathy results in a reduced number of 5-HT neurons in the raphe nucleus in both AD patients and mouse models [163,174,175]. Specifically, AD patients had significantly more neurofibrillary tangles in both MRN and DRN than in controls, leading to a 41% reduction in DRN neuron density and a 29% reduction in MRN neuron density [163]. Moreover, 2.6% of DRN neurons contain hyperphosphorylated Tau in AD patients at Braak stages 0 and 2, and hyperphosphorylated Tau in raphe nuclei deteriorate with AD progression [174]. Another study showed that neurofibrillary tangles were detected in the supratrochlear subnucleus of DRN in 1/5 of AD patients at Braak stage 0, and in all patients at Braak stages above 1 [175]. Consistent with AD patients, mice overexpressing human P301L Tau in the DRN recapitulate the degeneration of serotonergic system in AD patients, including profound 5-HT neuron loss, 5-HT neuron hyperexcitability, inflammation, axonal degeneration, as well as cognitive impairment and behavioral symptoms, including depressive- and anxiety-like behaviors, hyperlocomotion, and social deficits [172,176,177]. Upregulated inflammation in the DRN can exacerbate Tau phosphorylation, aggregation, and propagation alongside serotonergic projection targets [172]. It is also noteworthy that neurofibrillary tangles are thought to initiate from the DRN and then spread to its connected brain regions [176]. Taken together, these findings suggest that tauopathy, rather than that of Aβ pathology, is responsible for the degeneration of serotonergic system in AD.

### 7.2. Reduced 5-HT and Its Metabolites in the Hippocampus of AD Patients and Mouse Models

Accordingly, 5-HT and its metabolites were markedly altered in various brain regions of AD patients and mouse models (Table 2), including the hippocampus, cortex, amygdala, thalamus, substantia innominata, substantia nigra, putamen, and locus coeruleus [178,179]. Serotonergic deficiency seems to correlate with dementia type. For example, 5-HT levels were significantly increased in the brains of frontotemporal dementia (FTD) patients compared with AD patients, whereas the 5-HIAA/5-HT ratio was decreased in FTD patients compared with healthy controls [170]. In contrast, 5-HT and 5-HIAA were significantly reduced in 15 brain regions of AD patients comorbid with Down syndrome (DS) as compared to early-onset AD patients [180]. DS patients have monoaminergic deficits similar to DS + AD patients, suggesting that 5-HT deficiency is possibly due to early Aβ deposition [180]. Second, cerebral 5-HT levels were closely related to NPS in AD patients. Hippocampal 5-HIAA, 5-HIAA/5-HT, and 5-HIAA levels were significantly decreased in AD patients with aggression compared to those without aggression [171], and hippocampal 5-HIAA levels were negatively correlated with agitation scores in AD patients [181], suggesting that the brain-region specific alterations of 5-HT and its metabolites contribute to the occurrence of NPS in AD [171]. Lastly, the mini-mental state examination (MMSE) score, used as a measure of cognition, correlated positively with both hippocampal 5-HIAA levels and cortical 5-HT levels, indicating that hippocampal 5-HT/5-HIAA levels may predict cognitive decline in AD [181].

Consistent with AD patients, both 5-HT levels and 5-HIAA/5-HT ratio were significantly reduced in AD mouse models compared to age-matched controls. In the hippocampus of 18-month-old APP/PS1 mice, 5-HT was reduced by approximately 25% [182]. Similarly, hippocampal 5-HIAA/5-HT turnover was markedly lower in tgDimer mice than in non-transgenic mice [183]. Therefore, early Aβ deposition is associated with the progressive loss of hippocampal 5-HT with age [184]. Meanwhile, in a mouse model of co-expressing APP and Tau, 3×Tg-AD mice showed a significant decrease in 5-HT levels but a significant increase in 5-HIAA levels in the hippocampus [165]. It has been suggested that reduced hippocampal 5-HT and 5-HIAA levels are associated with increased depressive and aggressive behavior in 12-month-old THY-Tau22 mice [184].

**Table 2 ijms-26-01234-t002:** 5-HT and 5-HIAA levels in the hippocampus of AD patients and mouse models.

Research Model	Age (Year or Month)	Hippocampal 5-HT/5HIAA	References
MCI patient	73.2 ± 10.8 y	5-HT↓	[185]
MCI patient	83.0 ± 0 y	5-HIAA↓	[167]
AD patient	75.0 ± 11.9 y	5-HT↓, 5-HIAA↓	[186]
AD patient	70.0 ± 8.7 y	5-HT↓, 5-HIAA↓	[187,188]
AD patient	82.0 ± 8.0 y	5-HT↓, 5-HIAA↓	[181,189]
hTau mice	4 m	5-HT↓	[172]
hAPP-J20 mice	4–5 m	5-HT↓, 5-HIAA↓	[49]
5×FAD mice	9–10 m	5-HT↓	[190]
APPswe/PS1dE9 mice	2, 3m 4 m	5-HIAA (NC), 5-HT↓, 5-HIAA (NC)	[191]
3×Tg-AD mice	3 m	5-HT↓, 5-HIAA↑	[165]
THY-Tau22	12 m	5-HT↓, 5-HIAA↑	[184]
APP/PS1 mice	18 m	5-HT↓	[182]

Abbreviations: MCI: mild cognitive impairment; AD: Alzheimer’s disease; 5-HT: 5-hydroxytryptamine; 5-HIAA: 5-hydroxyindoleacetic acid; NC: no change; year or month: year or month; APP: amyloid precursor protein; PS1: presenilin-1; FAD: familial Alzheimer’s disease; Tg: transgenic; ↓ indicates a decrease in 5-HT or 5-HTIAA levels; ↑ indicates an increase in 5-HT or 5-HTIAA levels.

The 5-HT deficiency in AD patients can be attributed to decreased upstream production and increased downstream degradation. TPH2 activity was increased by 4.7-fold, and 5-HT and 5-HIAA levels increased by 4- and 2-fold, respectively, in the soma of raphe neurons in AD brains compared to healthy individuals; in contrast, the level of 5-HT and 5-HIAA decreased by 41% and 50%, respectively, in the synaptic terminals projecting to the amygdala [189]. However, the overt TPH protein or TPH-positive neurons were found to be significantly reduced in the DRN of hTau mice [192]. Another explanation for cerebral 5-HT deficiency is the reduced tryptophan levels in the cerebrospinal fluid (CSF) of AD patients [193]. Therefore, the hippocampal 5-HT deficiency may result from the diminished transport of TPH to axon terminals, the overt reduction in TPH protein, or metabolic alterations [192].

### 7.3. The Expression of 5-HTRs and SERT in the Hippocampus of AD Patients and Mouse Models

The 5-HT-dependent regulation of emotion and cognition relies on its binding to different 5-HTRs. According to their topographic structure, 5-HTRs are divided into two families with 14 subclasses: the first type belongs to G-protein coupled receptors (GPCRs), including 5-HT_1A_/BR and 5-HT_7_R (Gi-coupled), 5-HT_2A/B/C_R (Gq/11-coupled), 5-HT_4_R, 5-HT_6_R, and 5-HT_7_R (Gs-coupled) [13]. Another type is the ligand-gated cation channel, which comprises only 5-HT_3A_R. Previous functional tomographic imaging combined with immunohistochemical studies have shown that the expression of 5-HTRs is disrupted in the cerebral cortex, hippocampus, amygdala, and hypothalamus area of AD patients and mouse models; some receptors, such as 5-HT_6_Rs, have been summarized in other excellent reviews [14,194,195]. The main focus here is on the three highly expressed 5-HTRs (5-HT_1A_R, 5-HT_2A_R, 5-HT_3A_R, and 5-HT_4_R) in the hippocampus. Table 3 summarizes the expression of 5-HTRs in the hippocampus of AD patients and mouse models.

The 5-HT_1A_Rs are located either on the soma and dendrites of serotonergic neurons acting as autoreceptors, where their activation results in the inhibition of 5-HT release, or on the postsynaptic neurons acting as heteroreceptors. Positron emission tomography (PET) combined with isotopically labeled 5-HT_1A_R antagonists has shown that 5-HT_1A_Rs are significantly reduced in the hippocampi and raphe nuclei of AD patients and mouse models [49,196], and the reduction in hippocampal 5-HT_1A_Rs was positively correlated with cognitive decline while inversely correlated with increased amyloid plaque burden [197,198,199]. Interestingly, the hippocampal 5-HT_1A_Rs in MCI patients are much higher than in the healthy controls but lower than in mild AD patients, suggesting that hippocampal 5-HT_1A_Rs may distinguish MCI patients from mild AD patients [185]. The decreased 5-HT_1A_Rs can be interpreted as a consequence of CA1 neuron loss in the late stages of AD, which is supported by the evidence that hippocampal 5-HT_1A_Rs are preserved until the progression of neurofibrillary tangles in the late stages of AD [199,200]. Moreover, high levels of Aβ_40_ fibrils, but not Aβ_42,_ can induce the transient astrocytic overexpression of 5-HT_1A_Rs to compensate for hippocampal neuronal loss in AD [201]. Thus, whether the increased expression of 5-HT_1A_Rs in early AD is beneficial or detrimental is yet to be determined [194].

Additionally, there is a 20–30% reduction in cortical 5-HT_2A_Rs in MCI patients compared to healthy controls [202]. Intrahippocampal injection of aggregated Aβ_42_ in rats reduced hippocampal 5-HT_2A_Rs, and age-related amyloid burden was accompanied by a significant decrease in 5-HT_2A_R binding, suggesting a causal relationship between Aβ accumulation and the downregulation of hippocampal 5-HT_2A_Rs [203,204]. In contrast, 5-HT_2B_R mRNA levels were 2-fold higher in Aβ plaque-containing neocortex and hippocampus of old AD patients and mouse models than in age-matched controls [205].

PET imaging results showed that the density of 5-HT_3_R remains stable in the amygdala and hippocampus of AD patients compared to healthy individuals [206]. However, our previous results showed that the 5-HT_3_R mRNA started to decline in 4-month-old hAPP-J20 mice, while the 5-HT_3_R levels were significantly reduced in 8-month-old hAPP-J20 mice. Given that 5-HT_3_R mediates the tonic inhibition of pyramidal neurons, the reduction in hippocampal 5-HT_3_R contributes to the hyperexcitability of CA1 pyramidal neurons in hAPP-J20 mice [49]. Meanwhile, PET imaging results also showed a marked loss of 5-HT_4_R in the hippocampus and frontal cortex of AD patients [207], but there was no significant change in 5-HT_4_R in the DG of hAPP-J20 mouse models [49]. The discrepant results between AD patients and mouse models may be attributed to factors, such as experimental methods, as well as APP- or Tau-expressing genes’ age.

**Table 3 ijms-26-01234-t003:** The expression of 5-HTRs in the hippocampus of AD patients and mouse models.

Research Model	Age (Year or Month)	Hippocampal 5-HTRs	Reference
MCI patient	76.8 ± 11.6 y	5-HT_1A_R↓	[197]
MCI patient	83.0 ± 0 y	5-HT_1A_R↓	[208]
MCI patient	73.2 ± 10.8 y	5-HT_1A_R↑	[185]
MCI patient	73.2 ± 10.8 y	5-HT_1A_R↑	[187]
AD patient	81.7 ± 2.0 y	5-HT_1A_R↓	[208]
AD patient	75.0 ± 11.9 y	5-HT_1A_R↓	[197,199]
AD patient	70.0 ± 8.7 y	5-HT_1A_R↓	[185,187]
AD patient	82.0 ± 8.0 y	5-HT_4_R↓	[209]
AD patient	83.1 ± 5.8 y	5-HT_4_R↓	[207]
AD patient	77.0 ± 4.0 y	5-HT_3_R (NC)	[206]
HTau mice	4 m	5-HT_1B_R↓, 5-HT_4_R↑, 5-HT_1A_R↑, 5-HT_7_R↑	[172]
hAPP-J20 mice	4, 8 m	5-HT_1A_R↓, 5-HT_3A_R↓, 5-HT_4_R (NC)	[49]
5×FAD mice	3, 6 m	5-HT_1B_R↓, 5-HT_2B_R↓, 5-HT_3B_R↓, 5-HT_4_R↓, 5-HT_6_R↓, 5-HT_7_R↓, 5-HT_1F_R↓	[68]
APPswe/PS1dE9 mice	4, 8, 11 m 18 m	5-HT_2A_R (NC), 5-HT_2B_R↑	[203,205]
3×Tg-AD mice	24 m	5-HT_1A_R↓	[196]
Tg2576 mice	24 m	5-HT_1B_R↓	[210]
APP/PS1 mice	9 m	5-HT_1B_R↓	[211]

Abbreviations: MCI: mild cognitive impairment; AD: Alzheimer’s disease; 5-HT: 5- hydroxytryptamine; 5-HIAA: 5-hydroxyindoleacetic acid; NC: no change; y or m: year or month; APP: amyloid precursor protein; PS1: presenilin-1; FAD: familial Alzheimer’s disease; Tg: transgenic; 5-HTRs: 5-HT receptors; ↓ indicates a decrease in the 5-HTRs levels; ↑ indicates an increase in the 5-HTRs levels.

SERT is commonly used as a marker of serotonergic fibers. Reduced SERT availability has been observed in the corticolimbic regions of MCI patients compared to the controls [212]. Similarly, dementia patients with Lewy bodies have lower SERT binding in limbic brain regions than patients without dementia [213]. A reduction in SERT expression has also been found in Tg2576 mice at different ages [210]. These findings suggest that the reduced SERT levels may account for the disruption of 5-HT neurotransmission in AD patients.

### 7.4. Ascending Projections from Raphe to Hippocampus in AD

The monoaminergic neurons show degenerative changes in the early stages of AD, and it has previously been shown that the progression of Aβ deposition is accompanied by a progressive loss of monoaminergic afferents to the forebrain in APP/PS1 mice, and that axonal degeneration is probably due to the atrophy or loss of monoaminergic neurons [173]. However, our results showed that there was a significant increase in the serotonergic fibers projecting to the hippocampus in both AD patients and hAPP-J20 mice [49]. Anterograde and retrograde tracing showed that the aberrant serotonergic fibers projecting to the hippocampus mainly originated from the MRN^5-HT^ neuron. Importantly, no significant change was found between hAPP-J20 mice and non-transgenic mice in either DRN^5-HT^ or MRN^5-HT^ neurons, suggesting that the increased serotonergic sprouting in the hippocampus is not the result of raphe neuron loss [49]. Similarly, J. J. Rodriguez et al. showed that there was a significant increase in SERT fibers in the hippocampus of 3 and 18 month-old 3×Tg-AD mice, and the increased SERT fibers were specifically located in the SR/SLM border of CA1 region [165,214,215]. Thus, we and others have generally shown that serotonergic sprouting from raphe to the hippocampus is abnormal in AD patients and mouse models. In addition, glutamatergic afferents from MRN to the hippocampus are impaired by ETV4-dependent inhibition of vGluT3 transcription in APP/PS1 mice, and enhancing glutamatergic transmission to hippocampal PV interneurons improves the spatial memory retrieval in AD mice [23].

On the other hand, serotonergic projections from DRN to the hippocampus were also impaired in Aβ- or Tau-overexpressing mouse models. The retrograde tracing results showed that the number of Dil-labeled (retrograde tracer) cells was reduced by 44.6% in the DRN of 5×FAD mice compared to littermate controls, suggesting that the attenuated DRN–hippocampal serotonergic input may contribute to the decreased of 5-HT levels in the hippocampus of AD patients [216]. Another independent group also found that both the activity of DRN^5-HT^ neurons and their terminals projecting to the dorsal hippocampal CA1^CaMKII^ neurons were significantly reduced in the brain of 5×FAD mice [68]. Reduced SERT expression has also been observed in the DRN of MCI patients and is associated with lower resting-state hippocampal connectivity [217]. Notably, accumulation of phosphorylated Tau in the DRN leads to the reduced activity of DRN^5-HT^ neurons and the loss of serotonergic innervation of entorhinal cortex and hippocampal DG area, which may explain the depressive symptoms in late AD [172]. Reasonably, specific serotonergic denervation exacerbates Tau phosphorylation and cognition without affecting Aβ pathology in APP/PS1 mice [218]. Overall, tauopathy exerts inhibitory effects on 5-HT neurotransmission, mainly via inhibition of DRN^5-HT^ neurons.

### 7.5. Interactions Between Serotonergic and Glutamatergic Circuits in AD

Serotonergic and vGluT3^+^ neurons in the raphe nucleus differ widely in their projection pattern, electrophysiological properties, and modulation of hippocampal theta rhythm [47,56,58,75,87]; however, there is a small population of raphe neurons that co-express 5-HT and vGluT3 [41,56], and these vGluT3-positive serotonergic fibers show little or no SERT expression [21]. Notably, vGluT3 can regulate serotonergic transmission. The loss of vGluT3 in raphe neurons significantly reduces the 5-HT_1A_R-mediated auto-inhibition of serotonergic neurons [21]. On the other hand, vGluT3 can accelerate synaptic 5-HT transmission in the hippocampus [21]. However, these interactions are disrupted in the AD background. First, vGluT3 protein and mRNA levels are reduced in MRN^vGluT3^ neurons in 6-month-old APP/PS1 mice, which may contribute to the impairment of MRN^vGluT3^-DG^PV^ synaptic transmission and spatial memory retrieval in AD [23]. Restoration of vGluT3 expression in MRN^vGluT3^ neurons rescues the spatial memory deficits in APP/PS1 mice, whereas inhibition of vGluT3 transcription in MRN^vGluT3^ neurons exacerbates the spatial memory retrieval in APP/PS1 mice [23]. Taken together, loss of vGluT3 in the MRN impairs hippocampus-associated spatial memory retrieval in AD, possibly by decreasing 5-HT synaptic transmission.

On the other hand, glutamatergic hyperactivity in early AD may induce compensatory serotonergic fibers sprouting in the hippocampus. There is extensive serotonergic sprouting and increased APP expression in response to excitotoxic lesions, and compensatory serotonergic sprouting may counteract excitotoxicity via 5-HT-induced membrane hyperpolarization [219]. Our results have previously shown that abnormal serotonergic sprouting correlates with the hyperexcitability of CA1 pyramidal neurons in hAPP-J20 mice [49]. Specifically, reduced 5-HT/5-HT_3A_R and 5-HT/5-HT_1A_R signaling in the downstream synaptic terminals contribute to the hyperexcitability of CA1 pyramidal neurons in hAPP-J20 mice, suggesting that hippocampal serotonergic sprouting may compensate for 5-HT/5-HT_3A_R and 5-HT/5-HT_1A_R-induced hyperexcitability of excitatory neurons in early AD [49]. In addition, reduced DRN^5-HT^-CA1^CaMKII^ serotonergic transmission promotes hyperactivity of CA1^CaMKII^ neurons in 5×FAD mice, and activation of DRN^5-HT^ neurons ameliorates cognitive deficits in AD via 5-HT_1B_R/5-HT_4_R -mediated modulation of CA1^CaMKII^ neurons [68,211]. Overall, aberrant 5-HT signaling contributes to the hyperexcitability of hippocampal glutamatergic neurons in early AD.

## 8. Therapeutic Strategies to Rectify Serotonergic System for the Prevention of AD

### 8.1. Modulating 5-HTRs in the Treatment of AD

The first-line clinical drugs used to treat AD are cholinesterase inhibitors (ChEIs), such as donepezil, rivastigmine, and galantamine. The use of ChEIs has shown promising results in improving cognition, global status, and activities of daily living in patients with mild, moderate, or severe AD [2]. However, a meta-analysis showed that these drugs had limited effects in the treatment of NPS in AD patients [220]. Based on the above-mentioned causal relationship between aberrant 5-HTRs expression and NPS in AD, 5-HTRs can be seen as the therapeutic targets for the treatment of AD (Table 4).

Altered 5-HT_1A_R expression has been associated with depressive symptoms in AD. A 5-HT_1A_R partial agonist, tandospirone, has been shown to improve NPS, including delusions, agitation, depression, anxiety, and irritability, in dementia patients without causing severe adverse effects [221]. Meanwhile, application of a selective 5-HT_2C_R antagonist, RS-102221, can prevent Tau hyperphosphorylation and improve hippocampal LTP and spatial memory in stressed mice [222], suggesting that blockade of 5-HT_2C_R may attenuate cognitive deficits in AD. Additionally, 5-HT_2C_R polymorphism (102 T/C) is associated with an increased prevalence of delusional symptoms and treatment resistance to second-generation antipsychotics in AD patients [223]. Although no treatment has been approved by the United States Food and Drug Administration (FDA) for psychosis in AD, a new drug (pimavanserin) was approved by the FDA for the treatment of Parkinsion’s psychosis [224]. Pimavanserin, which served as a selective 5-HT_2A_R reverse agonist/antagonist, has no significant affinity for dopaminergic, muscarinic, or adrenergic receptors, offering hope that it may achieve positive results in the treatment of AD psychosis [224].

The 5-HT_4_Rs are highly expressed in the hippocampus, and their downstream signaling has been shown to modify APP processing [225]. Activation of 5-HT_4_Rs stimulates α-secretase processing, resulting in increased production of soluble forms of APP (sAPPα) and decreased Aβ levels, and this effect can be inhibited by 5-HT_4_R antagonists [225]. Moreover, a novel high-affinity 5-HT_4_R agonist, SSP-002392, can increase sAPPα production rate more than traditional 5-HT_4_R agonists, prucalopride, chronic administration of SSP-002392 reduced soluble and insoluble Aβ levels in the hippocampus of hAPP/PS1 mice [226]. Other 5-HT_4_R agonists, as well as 5-HT_6_R antagonists, have shown similar memory-improving effects in AD mouse models [227,228].

We have previously reported that 5-HT_1A_Rs and 5-HT_3A_Rs are downregulated in the hippocampus of hAPP-J20 mice, and stimulation of 5-HT_1A_R or 5-HT_3_R alone impairs contextual memory in wild-type mice without affecting memory performance in hAPP-J20 mice [49]. However, 5-HT_1A_R and 5-HT_3A_R agonists can work together to improve the contextual and spatial memory in hAPP-J20 mice [49], raising the possibility that a synergistic action of 5-HTRs may have a better performance in AD treatment.

**Table 4 ijms-26-01234-t004:** Targeting serotonergic or glutamatergic system for the prevention of AD.

Drugs	Target	Research Model	Results	Reference
Erythropoietin	5-HT_4_R, 5-HT_6_R, 5-HT_7_R, 5-HT_1A_R	C57BL/6 mice injected with Aβo	Erythropoietin ameliorates cognitive deficits in Aβo-induced AD mouse model by modulating 5-HTRs.	[229]
T3	SERT, 5HT_1A_R	3×Tg-AD mice	T3 supplements improve depression- like behavior in 3×Tg-AD mice via activation of 5HT_1A_R.	[230]
Paroxetine	SERT	APP/PS1 mice	Paroxetine treatment ameliorates motional dysfunction in APP/PS1 mice.	[231]
Desloratadine	5-HT_2A_R	APP/PS1 mice	Desloratadine represses Aβ level via upregulation of 5HT_2A_R-mediated Sirt1 expression and stimulation of autophagy.	[232]
Amisulpride	5-HT_7_R	Tau^P301L^-BiFC mice	Amisulpride mitigates Tauopathy via blockade of 5-HT_7_R activity.	[233]
Pimavanserin	5-HT_2A_R	APP/PS1 mice	Pimavanserin reduces Aβ levels via suppression of 5HT2A-R activity.	[234]
Riluzole	EAAT2/GLT-1	AβPP/PS1 mice	Riluzole benefits cognition in AβPP/PS1 mice by reducing glutamatergic tone.	[235]
Δ9-THC and CBD	GLT-1, EAAT_3_	APP/PS1 mice	Chronic combined treatment with Δ9-THC and CBD reduces hippocampal glutamate levels in APP/PS1 mice.	[236]
Decanoic acid	AMPAR	5×FAD mice	Decanoic acid improves cognitive function in 5×FAD mice by normalizing AMPAR-mediated signaling in CA1 hippocampal cells.	[237]
Perampanel	AMPAR	C57BL/6 mice injected with Aβ_1–42_	Perampanel restores Aβ-impaired hippocampal LTP and blocks Aβ-induced network hyperexcitability.	[238]
Troriluzole	vGlut1	3×Tg-AD	Troriluzole improves memory in 3×Tg mice by reducing amyloid, tau, vGlut1 levels and restoring glutamate, synaptic functions.	[239]

Abbreviations: MCI: mild cognitive impairment; AD: Alzheimer’s disease; 5-HT: 5-hydroxytryptamine; Aβ: beta amyloid; Aβo: Aβ oligomers; APP: amyloid precursor protein; PS1: presenilin-1; FAD: familial Alzheimer’s disease; Tg: transgenic; 5-HTRs: 5-HT receptors; Δ9-THC: Δ9-tetrahydrocannabinol; CBD: cannabidiol; NMDAR: N-methyl-D-aspartic acid receptor; EAAT: excitatory amino acid transporter; GLT: glutamate transporter; AMPAR: α-amino-3-hydroxy-5-methyl-4-isoxazolepropionic acid receptor; GlyT: glycine transporter; vGlut1: vesicular glutamate transporter 1.

### 8.2. SSRIs in the Treatment of AD

The loss of 5-HT release is associated with amyloid plaque deposition [164], while dietary tryptophan supplementation or the direct infusion of 5-HT into the hippocampus significantly reduced Aβ levels in the CA1 pyramidal neurons and brain interstitial fluid (ISF) through the Erk-dependent regulation of APP processing [15,215]. These findings suggest that 5-HT deficiency is implicated in APP processing and cerebral Aβ accumulation; therefore, 5-HT augmentation therapies are widely used in clinical trials for AD treatment. The most commonly used drugs to increase 5-HT are SSRIs, such as fluoxetine, paroxetine, escitalopram, fluvoxamine maleate, and sertraline. In the mouse models of AD, acute treatment with escitalopram reduced ISF Aβ by 25%, while chronic administration of escitalopram (5 mg/day) markedly decreased Aβ plaque load by 38% [16]. Another analog of escitalopram, citalopram, reduced brain plaque load in AD mice by as much as 50% [15]. Additionally, paroxetine treatment not only reduced intraneuronal Aβ levels, but also reduced Tau pathology in male 3×Tg-AD mice [240]. In line with these results, short-term escitalopram treatment induces a 9.4% reduction in CSF Aβ_42_ in cognitively normal older adults compared to placebo groups [241]. In particular, prodromal AD patients exposed to long-term antidepressants drugs (≥5 years) had a significantly lower amyloid burden than those not exposed [15]. Compared with short-term SSRI treatment in MCI patients with depression symptoms, long-term SSRI treatment (>5 years) delayed the progression from MCI to AD by approximately 3 years [17].

In contrast to the beneficial effects of SSRIs, several independent groups have found that SSRIs have minimal effects on mitigating Aβ pathology in AD mouse models [242,243,244]. For example, three months of 5-HT depletion had no effect on reducing Aβ plaques and Aβ_42_/Aβ_40_ ratios in 12-month-old APP/PS1 mice [242], nor did TPH2 inactivation or serotonergic deafferentation, which exacerbate Aβ pathology [244,245]. Concomitantly, increasing cerebral 5-HT levels by chronic paroxetine treatment did not reduce the number and size of Aβ plaques in 18-month-old APP/PS1 mice [242,243]. Even worse, chronic paroxetine treatment not only failed to mitigate Aβ pathology, but also increased mortality in APP/PS1 mice [244]. These results suggest that chronic SSRI treatment does not reverse Aβ pathology.

Overall, in agreement with the view of J.J Rodriguez [14], the discrepant outcomes of SSRI treatment need to take into account the heterogeneity of experimental design, e.g., treatment duration, species and age of experimental models, history of drug use, number of animals or patients enrolled, and drug dosage.

### 8.3. Stimulation of Raphe Nuclei in the Treatment of AD

Given that MR neurons can regulate hippocampal ripple activity and memory consolidation [99,134], stimulation of raphe neurons is considered a novel strategy to prevent the development of AD. Stimulation of midbrain MRN and DRN has been tested for its ability to regulate recovery from traumatic brain injury (TBI), with results showing that 8 Hz stimulation of the MRN not only restored the reference memory in the Morris water maze, but also markedly reduced the cortical neuronal loss in TBI mice [246]. Mechanistically, repeated electroconvulsive stimulation enhanced serotonergic sprouting and increased 5-HT metabolism in the partially lesioned hippocampus [247]. Electroacupuncture at Shenshu and Baihui sites effectively improved synaptic plasticity and memory in AD via activation of DRN^5-HT^ neurons, whereas chemogenetic inhibition of DRN^5-HT^ neurons abolished the beneficial effects of electroacupuncture in APP/PS1 mice [211]. These results remind us that stimulation of raphe nuclei holds great potential in the treatment of AD.

We have previously shown that chemogenetic activation of MRN^5-HT^ neurons improves memory in hAPP-J20 mice via 5-HT_3A_R- and 5-HT1AR-mediated inhibition of hyperexcitability of CA1 pyramidal neurons [49]. In alignment with our findings, Chen et al. found that stimulation of DRN^5-HT^ neurons attenuated depressive symptoms and cognitive impairment in 5×FAD mice via 5-HT_1B_R and 5-HT_4_R-mediated activation of dorsal CA1 ^CaMKII^ neurons [68]. He et al. showed that activation of MRN^vGluT3^ neurons rescued spatial memory deficits in APP/PS1 mice by activating DG^PV^ interneurons [23]. Taken together, these results provide solid evidence that stimulation of raphe nuclei could be a promising therapeutic target for AD treatment.

## 9. Concluding Remarks

Previous studies have extensively elaborated the roles of raphe–hippocampal serotonergic circuits in the regulation of emotional and cognitive behaviors [10,14,158,248]. However, compared to 5-HT neurons, the role of raphe glutamatergic neurons in modulating hippocampal activity has received little attention. Notably, glutamatergic neurons within the raphe nuclei markedly outnumber 5-HT neurons in their innervation of the hippocampus, and in some instances, these glutamatergic neurons may exert opposing effects in modulating hippocampal activity. In addition, almost all AD patients suffer from BPSD throughout the disease, and there are currently no FDA-approved drugs for the treatment of psychosis in AD. Given that the disrupted raphe–hippocampal serotonergic and glutamatergic circuits are closely associated with the NPS and cognitive deficits in AD, SSRIs remain the first-line options for the management of NPS, but concerns have been raised regarding the potential adverse effects and abuse. It is encouraging that agomelatine, a new type of antidepressant, has shown promising results in alleviating the BPSD [249], so there is an urgent clinical need for new drugs and techniques without side effects to treat NPS in the early stages of AD.

## Figures and Tables

**Figure 1 ijms-26-01234-f001:**
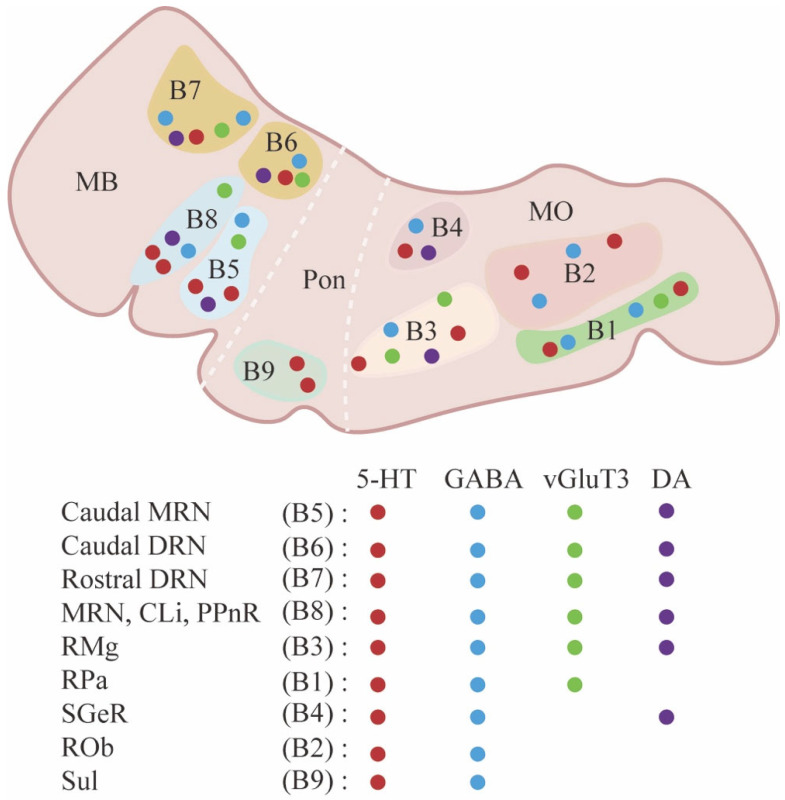
Anatomical and neurochemical diversity of raphe nuclei. Abbreviations: Caudal MRN: caudal median raphe nucleus; Caudal DRN: caudal dorsal raphe nucleus; Rostral DRN: rostral dorsal raphe nucleus; CLi: caudal linear nucleus; PPnR: prepontine raphe nucleus; RMg: raphe magnus nucleus; Rpa: raphe pallidus nucleus; SGeR: supragenual raphe nucleus; ROb: raphe obscurus nucleus; Sul: supralemniscal raphe complex; MB: midbrain; Pon: pons; MO: medulla oblongata.

**Figure 2 ijms-26-01234-f002:**
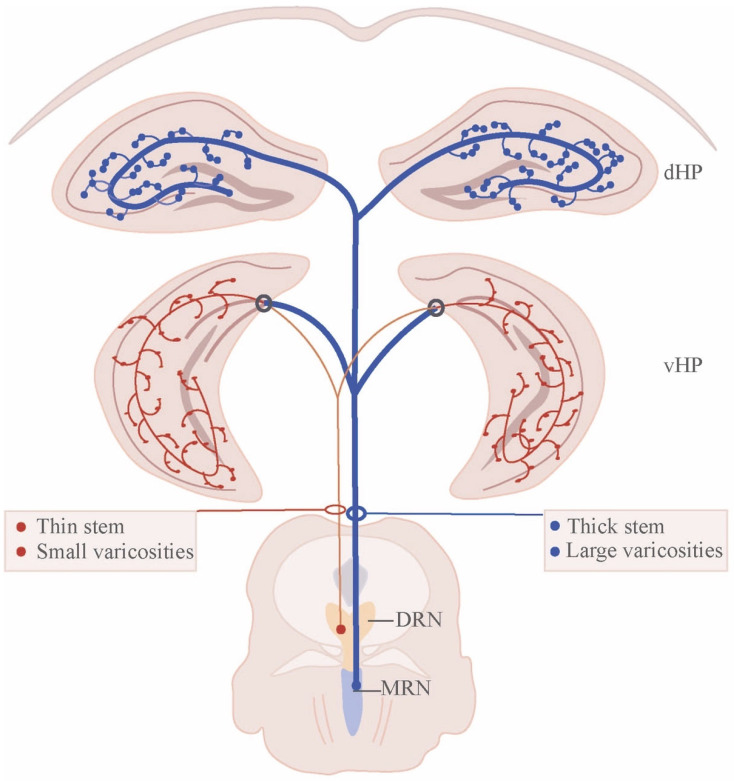
Innervation of dorsal and ventral hippocampus by the raphe nucleus. MRN mainly innervates the dorsal and ventral hippocampus. MRN-derived serotonergic neurons send beaded fibers, whereas DRN-derived neurons mainly innervate the vHP, and send regularly spaced fine fibers [36]. Abbreviations: dHP: dorsal hippocampus; vHP: ventral hippocampus; DRN: dorsal raphe nucleus; MRN: medial raphe nucleus.

**Figure 3 ijms-26-01234-f003:**
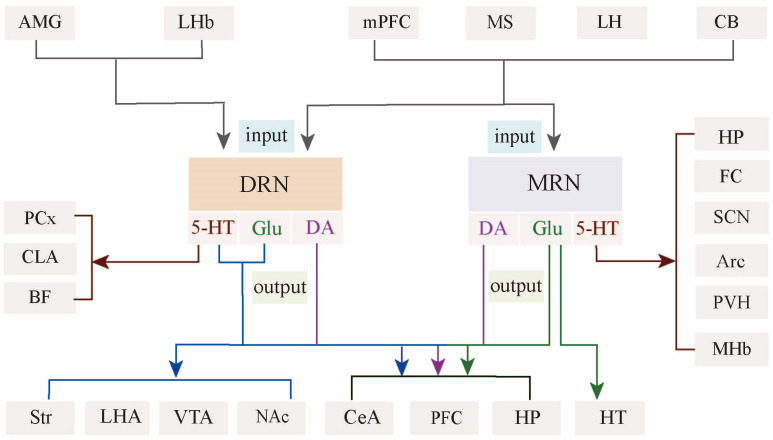
The efferent and afferent fibers of MRN correspond to midline structures, whereas those of DRN correspond to lateral structures. Both MRN and DRN innervate CeA, PFC, and HP. Abbreviations: DRN: dorsal raphe nucleus; MRN: medial raphe nucleus; AMG: amygdala; LHb: lateral habenula; mPFC: medial prefrontal cortex; MS: medial septal area; LH: lateral hypothalamus; CB: cerebellum; Pcx: piriform cortex; CLA: claustrum; BF: basal forebrain; Str: striatum; LHA: lateral hypothalamic area; VTA: ventral tegmental area; NAc: nucleus accumbens; CeA: central amygdala; PFC: prefrontal cortex; HP: hippocampus; HT: hypothalamus; FC: frontal cortex; SCN: suprachiasmatic nucleus; Arc: arcuate nucleus; PVH: paraventricular hypothalamic nucleus; MHb: medial habenula.

**Figure 4 ijms-26-01234-f004:**
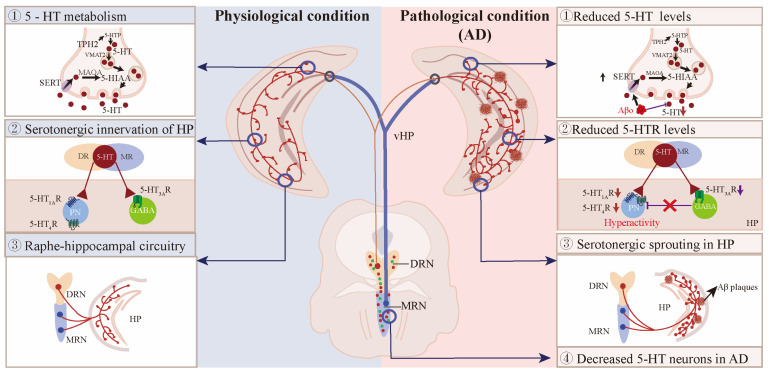
A schematic diagram for comparison of raphe–hippocampal circuit under physiological and pathological conditions (AD). Abbreviations: dHP: dorsal hippocampus; vHP: ventral hippocampus; DRN: dorsal raphe nucleus; MRN: medial raphe nucleus; SERT: serotonin transporter; AD: Alzheimer’s disease; 5-HT: 5- hydroxytryptamine; 5-HIAA: 5-hydroxyindoleacetic acid; 5-HTRs: 5-HT receptors; PN: pyramidal neuron; GABA: gamma-aminobutyric acid.

**Table 1 ijms-26-01234-t001:** The number of 5-HT neurons in AD patients and mouse models.

Sample Type	Age (Year or Month)	Measurement Method	Raphe Neurons	Reference
MCI patient	73.2 ± 10.8 y	Biochemical Detection	DRN^5-HT^↓	[166]
MCI patient	83.0 ± 0 y	HPLC	DRN^5-HT^↓, MRN^5-HT^↓	[167]
AD patient	70.0 ± 8.7 y	Biochemical Detection	DRN^5-HT^↓	[166]
AD patient	83.1 ± 5.8 y	ICC, 2DIA	MRN^5-HT^↓	[168]
AD patient	79.3 ± 8.7 y	QAR	DRN^5-HT^↓	[169]
AD patient	82.0 ± 1.0 y	ICC	DRN^5-HT^↓	[163]
AD patient	76.5 ± 10.2 y	RP-HPLC	DRN^5-HT^↓, MRN^5-HT^↓	[170]
AD patient	75.9 ± 7.3 y	RP-HPLC	DRN^5-HT^↑	[171]
hTau mice	4 m	IF	DRN^5-HT^↓	[172]
hAPP-J20 mice	4 m	IF, IHC	NC	[49]
3×Tg-AD mice	3–18 m	IHC	NC	[165]
5×FAD mice	3 m	IF	DRN^5-HT^↓	[68]
APPswe/PS1dE9 mice	24 m	IHC	DRN^5-HT^↓	[173]

Abbreviations: MCI: mild cognitive impairment; AD: Alzheimer’s disease; 5-HT: 5-hydroxytryptamine; 5-HIAA: 5-hydroxyindoleacetic acid; DR: dorsal raphe nucleus; MR: median raphe; NC: no change; y or m: year or month; IF: immunofluorescence; IHC: immunohistochemistry; RP-HPLC: reversed-phase high-performance liquid chromatography; ICC: immunocytochemistry; QAR: quantitative autoradiography; 2DIA: two-dimensional image analysis; ↓ indicates a decrease in the number of 5-HT neurons; ↑ indicates an increase in the number 5-HT neurons.

## Data Availability

Data will be made available on request.

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
