# Peer review of "Deciphering the Functions of Raphe–Hippocampal Serotonergic and Glutamatergic Circuits and Their Deficits in Alzheimer’s Disease"

_ijms, 2025, doi:10.3390/ijms26031234_

Round 1

Reviewer 1 Report

Comments and Suggestions for Authors

The authors summarized the anatomical, neurochemical, and electrophysiological diversity of the raphe nuclei and the raphe-hippocampal circuitry. They highlighted the role of raphe nuclei in regulating hippocampal activity, emotion, and cognition, and reviewed the disruption of these circuits in Alzheimer’s disease (AD) pathogenesis. Additionally, they analyzed potential therapies for alleviating neuropsychiatric symptoms and cognitive decline in AD. While the manuscript demonstrates novelty, the following issues need to be addressed:

·  Recommendation to add a new section: Discuss the interactions between raphe-hippocampal serotonergic and glutamatergic circuits in AD. Currently, these two circuits are presented as independent, with little connection between them.

·  Strengthen the connection: The manuscript should provide a more detailed analysis and discussion of the mechanisms underlying the interaction between raphe-hippocampal serotonergic and glutamatergic circuits in AD.

·  Suggestions for Table 1: It is recommended to include "Sample Type" and the "Measurement Method" for determining the number of 5-HT neurons in Table 1.

·  Expand on the topic mentioned in the title: The title refers to the "functions of raphe-hippocampal serotonergic and glutamatergic circuits." Consider adding a table summarizing the current therapeutic drugs that act by modulating the functions of the raphe-hippocampal serotonergic and glutamatergic circuits and explaining their mechanisms of action.

·  Correction of author attribution: The last author’s name seems to be incorrectly listed as “and.” Please correct this error.

Comments on the Quality of English Language

The authors summarized the anatomical, neurochemical, and electrophysiological diversity of the raphe nuclei and the raphe-hippocampal circuitry. They highlighted the role of raphe nuclei in regulating hippocampal activity, emotion, and cognition, and reviewed the disruption of these circuits in Alzheimer’s disease (AD) pathogenesis. Additionally, they analyzed potential therapies for alleviating neuropsychiatric symptoms and cognitive decline in AD. While the manuscript demonstrates novelty, the following issues need to be addressed:

·  Recommendation to add a new section: Discuss the interactions between raphe-hippocampal serotonergic and glutamatergic circuits in AD. Currently, these two circuits are presented as independent, with little connection between them.

·  Strengthen the connection: The manuscript should provide a more detailed analysis and discussion of the mechanisms underlying the interaction between raphe-hippocampal serotonergic and glutamatergic circuits in AD.

·  Suggestions for Table 1: It is recommended to include "Sample Type" and the "Measurement Method" for determining the number of 5-HT neurons in Table 1.

·  Expand on the topic mentioned in the title: The title refers to the "functions of raphe-hippocampal serotonergic and glutamatergic circuits." Consider adding a table summarizing the current therapeutic drugs that act by modulating the functions of the raphe-hippocampal serotonergic and glutamatergic circuits and explaining their mechanisms of action.

·  Correction of author attribution: The last author’s name seems to be incorrectly listed as “and.” Please correct this error.

Author Response

Point-by-point Response to Reviewer’s Comments

First of all, we thank the editorial board and the two reviewers very much for your constructive comments on our manuscript. Below is our response (shown as italic blue characters) to the comments.

Reviewer 1

The authors summarized the anatomical, neurochemical, and electrophysiological diversity of the raphe nuclei and the raphe-hippocampal circuitry. They highlighted the role of raphe nuclei in regulating hippocampal activity, emotion, and cognition, and reviewed the disruption of these circuits in Alzheimer’s disease (AD) pathogenesis. Additionally, they analyzed potential therapies for alleviating neuropsychiatric symptoms and cognitive decline in AD. While the manuscript demonstrates novelty, the following issues need to be addressed:

We thank the reviewer’s comments and valuable suggestions on our manuscript. Meanwhile, we carefully revised the English writing in the manuscript.

Comment 1: Recommendation to add a new section: Discuss the interactions between raphe-hippocampal serotonergic and glutamatergic circuits in AD. Currently, these two circuits are presented as independent, with little connection between them.

Response 1: Thanks for your suggestion, we supplemented a new section to discuss the interactions of serotonergic and glutamatergic circuits in AD in revised manuscript (Page 18 line 730-760).

Although serotonergic and vGluT3+ neurons in raphe nucleus differ widely in their projection pattern, electrophysiological properties, and modulation of hippocampal theta rhythm [47, 56, 58, 75, 87], however, there is a small population of raphe neurons that co-express 5-HT and vGluT3 [41, 56], and these vGluT3-positive serotonergic fibers show little or no SERT expression [21]. Notably, vGluT3 can regulate the serotonergic trans-mission. Loss of vGluT3 in raphe neurons significantly reduces 5-HT1AR-mediated in-hibition of serotonergic neurons [21]. On the other hand, vGluT3 can accelerate synaptic 5-HT transmission in hippocampus [21]. However, these interactions are disrupted in AD background. First, vGluT3 protein and mRNA levels are reduced in MRNvGluT3 neurons in 6-month-old APP/PS1 mice, which may contribute to the impairment of MRNvGluT3-DGPV synaptic transmission and spatial memory retrieval in AD [23]. Resto-ration of vGluT3 expression in MRNvGluT3 neurons rescues the spatial memory deficits in APP/PS1 mice, whereas inhibition of vGluT3 transcription in MRNvGluT3 neurons exac-erbates the spatial memory retrieval in APP/PS1 mice [23]. Taken together, loss of vGluT3 in MRN impairs hippocampus-associated spatial memory retrieval in AD, possibly by decreasing 5-HT synaptic transmission.

On the other hand, glutamatergic hyperactivity in early AD may induce com-pensatory serotonergic fibers sprouting in the hippocampus. There is extensive sero-tonergic sprouting and increased APP expression in response to excitotoxic lesion, and the compensatory serotonergic sprouting may counteract the excitotoxicity via 5-HT-induced membrane hyperpolarization [224]. Our results have previously shown that abnormal serotonergic sprouting correlates with hyperexcitability of CA1 pyramidal neurons in hAPP-J20 mice [49]. Specifically, reduced 5-HT/5-HT3AR and 5-HT/5-HT1AR signaling in the downstream synaptic terminals contributes to hyperexcitability of CA1 pyramidal neurons in hAPP-J20 mice, suggesting that hippocampal serotonergic sprouting may compensate for 5-HT/5-HT3AR and 5-HT/5-HT1AR-induced hyperexcitability of excitatory neurons in early AD [49]. In addition, reduced DRN5-HT-CA1CaMKâ…¡serotonergic trans-mission promotes hyperactivity of CA1CaMKâ…¡ neurons in 5×FAD mice, and activation of DRN5-HT neurons ameliorates cognitive deficits in AD via 5-HT1BR/5-HT4R -mediated modulation of CA1CaMKâ…¡ neurons [68, 216]. Overall, aberrant 5-HT signaling contributes to the hyperexcitability of hippocampal glutamatergic neurons in early AD.

Comment 2: Strengthen the connection: The manuscript should provide a more detailed analysis and discussion of the mechanisms underlying the interaction between raphe-hippocampal serotonergic and glutamatergic circuits in AD.

Response 2: We agree with the reviewer’s opinion that interaction between raphe-hippocampal and glutamatergic circuits in AD need to be discussed in detail. We carefully searched these interactions in PubMed and Web of Science, and discussed these interactions in new section (entitled 6.5. Interactions between serotonergic and glutamatergic circuits in AD) (Page 18 line 730-760).

Comment 3: Suggestions for Table 1: It is recommended to include "Sample Type" and the "Measurement Method" for determining the number of 5-HT neurons in Table 1.

Response 3: Thanks for your suggestions. We have revised the Table 1 in the submitted manuscript. Accordingly, we also revised the Table 2 and 3 to be consistent with Table1.

Comment 4: Expand on the topic mentioned in the title: The title refers to the "functions of raphe-hippocampal serotonergic and glutamatergic circuits." Consider adding a table summarizing the current therapeutic drugs that act by modulating the functions of the raphe-hippocampal serotonergic and glutamatergic circuits and explaining their mechanisms of action.

Response 4: That’s a good suggestion, we added a new table (Table 4) to summarize the current therapeutic drugs targeting at serotonergic and glutamatergic circuits for the prevention of AD.

Comment 5: Correction of author attribution: The last author’s name seems to be incorrectly listed as “and.” Please correct this error.

Response 5: We apologize for the inconvenience in your reading, we have revised the author affiliations in the manuscript.

Reviewer 2 Report

Comments and Suggestions for Authors

Alzheimer's disease (AD) presents a significant challenge for modern science and healthcare, with its prevalence projected to rise due to increasing life expectancy. Current treatments remain limited in their effectiveness. The subcortical innervation of the hippocampus by the raphe nucleus plays a vital role in emotional and cognitive regulation. Recent research has uncovered a strong connection between AD and disruptions in the serotonergic and glutamatergic circuits of the raphe-hippocampal pathway, highlighting this as a promising therapeutic target.

This comprehensive review titled “Deciphering the functions of raphe-hippocampal serotonergic and glutamatergic circuit and its deficits in Alzheimer’s disease” outlines the latest findings regarding the cellular and functional aspects of the raphe-hippocampal circuit in both healthy and pathological states. Additionally, it explores therapeutic strategies that leverage this circuit to alleviate neuropsychiatric symptoms and cognitive decline in AD. The detailed discussion on serotonergic pathways offers valuable insights, serving as a resource for generating new hypotheses and identifying potential drug targets. Overall, I think this is a well-written review with sufficient references to support the author’s points. I have the following suggestion to improve the manuscript prior to its publication:

           Adding a schematic figure to compare the raphe-hippocampal circuit under physiological and pathological conditions would assist readers in visualizing and synthesizing the discussed concepts.

Author Response

Point-by-point Response to Reviewer’s Comments

First of all, we thank the editorial board and the two reviewers very much for your constructive comments on our manuscript. Below is our response (shown as italic blue characters) to the comments.

Reviewer 2
Alzheimer's disease (AD) presents a significant challenge for modern science and healthcare, with its prevalence projected to rise due to increasing life expectancy. Current treatments remain limited in their effectiveness. The subcortical innervation of the hippocampus by the raphe nucleus plays a vital role in emotional and cognitive regulation. Recent research has uncovered a strong connection between AD and disruptions in the serotonergic and glutamatergic circuits of the raphe-hippocampal pathway, highlighting this as a promising therapeutic target.

This comprehensive review titled “Deciphering the functions of raphe-hippocampal serotonergic and glutamatergic circuit and its deficits in Alzheimer’s disease” outlines the latest findings regarding the cellular and functional aspects of the raphe-hippocampal circuit in both healthy and pathological states. Additionally, it explores therapeutic strategies that leverage this circuit to alleviate neuropsychiatric symptoms and cognitive decline in AD. The detailed discussion on serotonergic pathways offers valuable insights, serving as a resource for generating new hypotheses and identifying potential drug targets. Overall, I think this is a well-written review with sufficient references to support the author’s points. I have the following suggestion to improve the manuscript prior to its publication:

We thank the reviewer’s comments and valuable suggestions on our manuscript.

    Comment 1: Adding a schematic figure to compare the raphe-hippocampal circuit under physiological and pathological conditions would assist readers in visualizing and synthesizing the discussed concepts.

Response 1: Thanks for your good suggestion, this, in deed, can help us to increase the visualization and readability of our paper. Therefore, we added a new Figure 4 in our revised manuscript (Page 13).